# Baroque Gardens in Transylvania: A Historic Overview

**Albert Fekete * and Máté Sárospataki**

Department of Garden Art and Landscape Design, Institute of Landscape Architecture, Urban Planning and Garden Art, Hungarian University of Agriculture and Life Sciences, 2100 Gödöllő, Hungary; sarospataki.mate@uni-mate.hu

\* Correspondence: fekete.albert@uni-mate.hu

**Abstract:** For over more than 20 years, Transylvanian ensembles, gardens and parks have been investigated, described and analysed by a research group from Hungary, led by Albert Fekete. The goal of this study of Transylvanian ensembles is to get background information, insight for developing a strategy for landscape preservation and development in the long run that comprises the cultural and historical values and the demands from society on what to do with them in the contemporary context. The goal of the article is to give an overview of what is already known and what could be done from the viewpoint of protection, planning and design. The research methods are mixed, but are largely based on the case study approach, supplemented by experimental design, fieldwork and research by design. The conclusion is that, given the state of what is left over from these historical artefacts, restoration in the strict sense will be impossible. This will be a major challenge for landscape architecture to take into account the historical values, integrate them with new functions and use and the recent demands of improving water management, energy transition and the creation of comfort and healthy living environments for people.

**Keywords:** history of gardens; castle garden; goosefoot avenue; star-shaped garden layout; Austro-Hungarian Empire; Transylvania as part of Romania

## 1. Introduction

The Carpathian Mountains are not only geographically but also culturally a crucial part of Europe's history. The multi-confessional zones along the fault lines of the Orthodox–Catholic (east–west) schism of 1054 and the Protestant–Catholic (north–south) schism of 1517 meet in the Carpatho–Pannonian region [1]. Regarding its geography, the Carpathian Basin is a coherent entity, while it is one of the most characteristic transition areas in Europe in terms of its spatial structure from political, economic and religious aspects [2]. In this landscape and regional context, Hungary and its neighbouring countries are interdependent not only geographically but also historically and culturally. For centuries, Hungary has played a leading role in this coexistence. From the conquest to the present day, the history of Transylvania is linked to the history of Hungary. For centuries, it existed either as part of Hungary or as an independent principality closely linked to Hungary, and in broad cultural and historical terms it can be interpreted as the easternmost bastion of Europe. The national unity, the intertwined history and the context of the single landscape unit also make it clear that the research of Hungarian garden art can and should be carried out comprehensively, covering the entire Carpathian Basin, and that the Transylvanian gardens from the modern period can be interpreted as some of the easternmost sites of the European garden styles from the 17th century onwards. In the history of Transylvanian gardens, the 18th century Baroque gardens that followed the late Renaissance gardens were mainly developed in connection with the residences of the local aristocracy and the ecclesiastical centres of the Roman Catholic Church (bishoprics, major monasteries). Accordingly, the main goal of the present study is to focus on research of the Baroque castle gardens of the region, highlighting their most important compositional aspects. Accordingly, the research

summarises, classifies and introduces the types of the Baroque gardens surveyed through specific examples, providing information on their role and significance in Hungarian and European garden art.

## 2. Historical Background

After the conquest (896), Hungarian history can be divided into several periods and this division is important for the historical, economic and social aspects of the region. Figure 1 shows the most important historical periods of the Kingdom of Hungary and, in this context, of Transylvania. According to this historical division, until the collapse of the Austro-Hungarian Empire the most significant and prosperous period of the kingdom was the third period (Figure 1c), which started after the Mongol Invasion of 1241, and lasted until the Battle of Mohács in 1526. During this period, the medieval Kingdom of Hungary became a regional power; its position in international policy was the most favourable possible and its internal legislative and administrative reforms reflected a fair spirit and a strong sense of justice [3–5].

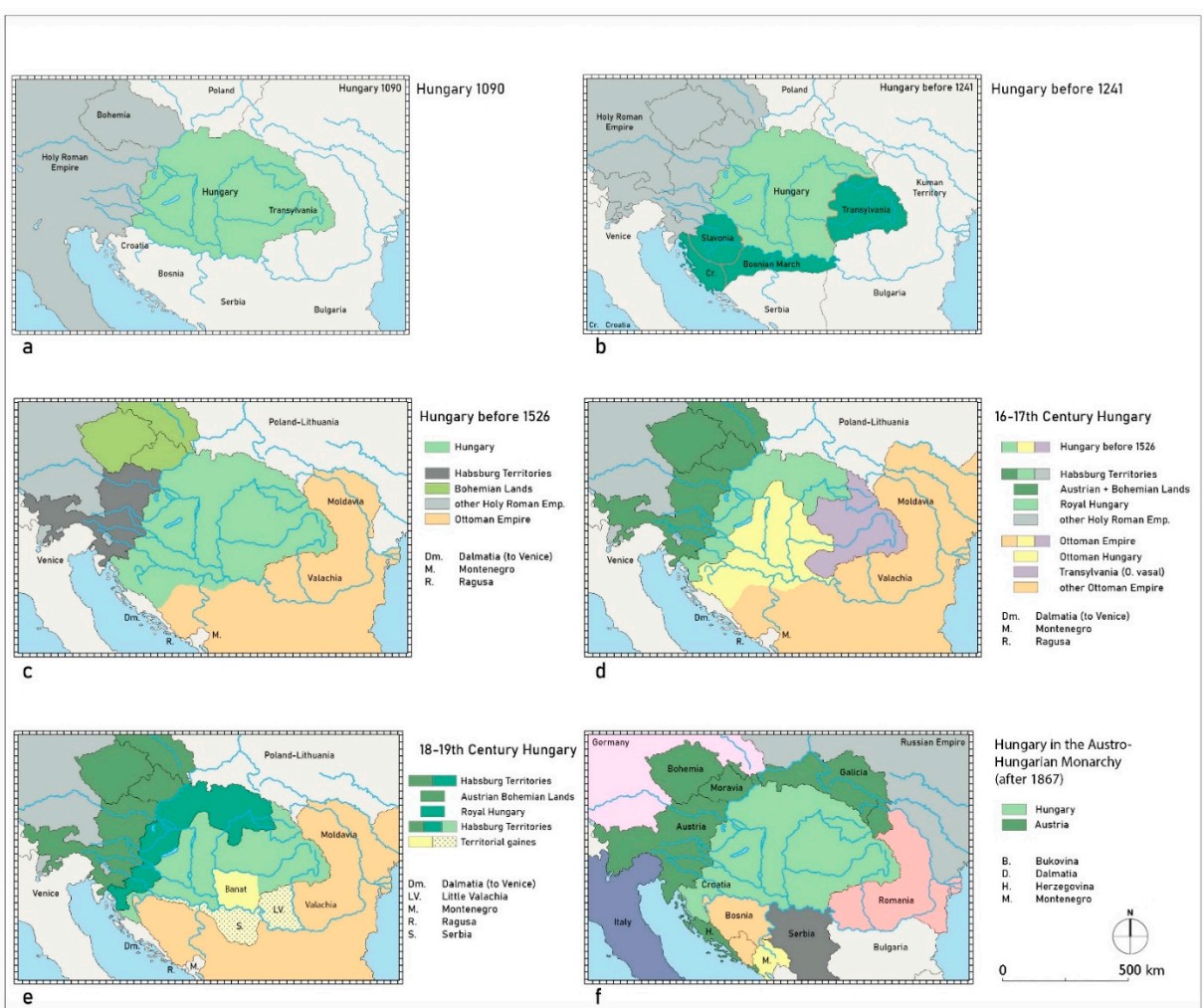

**Figure 1.** Graphical representation of the major historical periods of the Kingdom of Hungary. Source: prepared by the authors.

The Battle of Mohács, which resulted in an Ottoman victory, caused a major rupture not only in the history of Eastern Europe but also in its culture and landscape history. After the Ottoman rule following the Battle of Mohács, for a hundred and fifty years Western Hungary, Upper Hungary and Transylvania became the last bastions of the spiritual and material heritage and continuity of the European Christian culture in the region.

The fourth stage of the periods shown in Figure 1 includes the "golden age" of the independent Principality of Transylvania, 1613–1683, when Transylvania, although a vassal of the Porte, was economically prosperous, strengthened and flourishing [6–9]. During this period, Transylvanian garden culture in the late Renaissance reached a high level of excellence. At least 60 important Renaissance gardens are authentically mentioned and/or described in archival sources and in some of them it is still possible to detect Renaissance monuments and garden elements of the period [10–15].

The fifth historical period is particularly significant for the present study, as the period of absolutism following the final defeat of the Rákóczi War of Independence in 1711 marked the emergence and completion of the Baroque period in Hungary and Transylvania. From 1712, the supreme governing authority of Transylvania, the administrative and the judicial body of the region was the "Gubernium", directly subordinate to the Viennese court. Under the new balance of forces, Transylvania lost its autonomy and played a strongly subordinate role in the Habsburg Empire, both economically and politically, until the Compromise of 1867. The battles with the Ottomans or the Habsburgs, the political dependency of a once prosperous and powerful Transylvania in the 17th century, the division of the Transylvanian aristocracy and the decline of economic capabilities all contribute to the fact that the 18th century can be considered one of the darkest periods in Transylvanian history [16–19].

## 3. Materials and Methods

### 3.1. Timelines of Key European Garden Styles

When examining the characteristics of Transylvanian Baroque garden art, it is not only necessary to know the geographical and spatial location of the area under study as well as its historical background, but it is also important to know the exact chronological boundaries of the subject under study: the Baroque period of garden history. This temporal definition should be done in a European context [20–31], taking into consideration the well-known baroque garden examples and studying their compositional and functional characteristics. In this aspect the gardens from Versailles (FR), Vaux le Vicomte (FR), Montalto (IT), Hampton Court (UK), Het Loo (NL), Frederiksborg (DK), Herrenhausen (D), Nymphenburg (D), Schönbrunn (AU), Peterhof (RU) represent a comprehensive and relevant European pattern.

The development of gardens is always closely linked to the historical and economic development of a country or region. The analysis of the comparative table (Figure 2) shows that, until the national tragedy of Mohács, Hungarian garden art was on a par with European gardens. However, after the Battle of Mohács, during the 150 years of Ottoman occupation, it fell back considerably, and even after the expulsion of the Ottomans it took a century and a half to catch up the cultural delay. In the 19th century it came close to the ideas of Western European garden culture, but in terms of its overall spectrum it was only after the economic boom following the Compromise that it managed to catch up with this time lag that originated from the 16th century history [32,33]. Accordingly, Fatsar links the emergence of Baroque garden art to the appearance of the goosefoot-pattern avenues and the gradual abandonment of the orthogonal layout, which occurred at the turn of the 16th and 17th centuries [34] (Table 1).

In comparison with French or German Baroque, the transition from Renaissance to Baroque in Transylvania is much longer, and the Baroque style spread here about a hundred years later in the middle and second half of the 18th century. By this time, the connection between Western civilisation and Transylvania, which led through Hungary, had opened up. With the spread of the imperial Baroque, several western European architects and sculptors employed initially in Austria were invited to Transylvania by local aristocracy and as a result Transylvanian 18th-century architecture became an integral and representative part of the Baroque art of the monarchy [35,36].

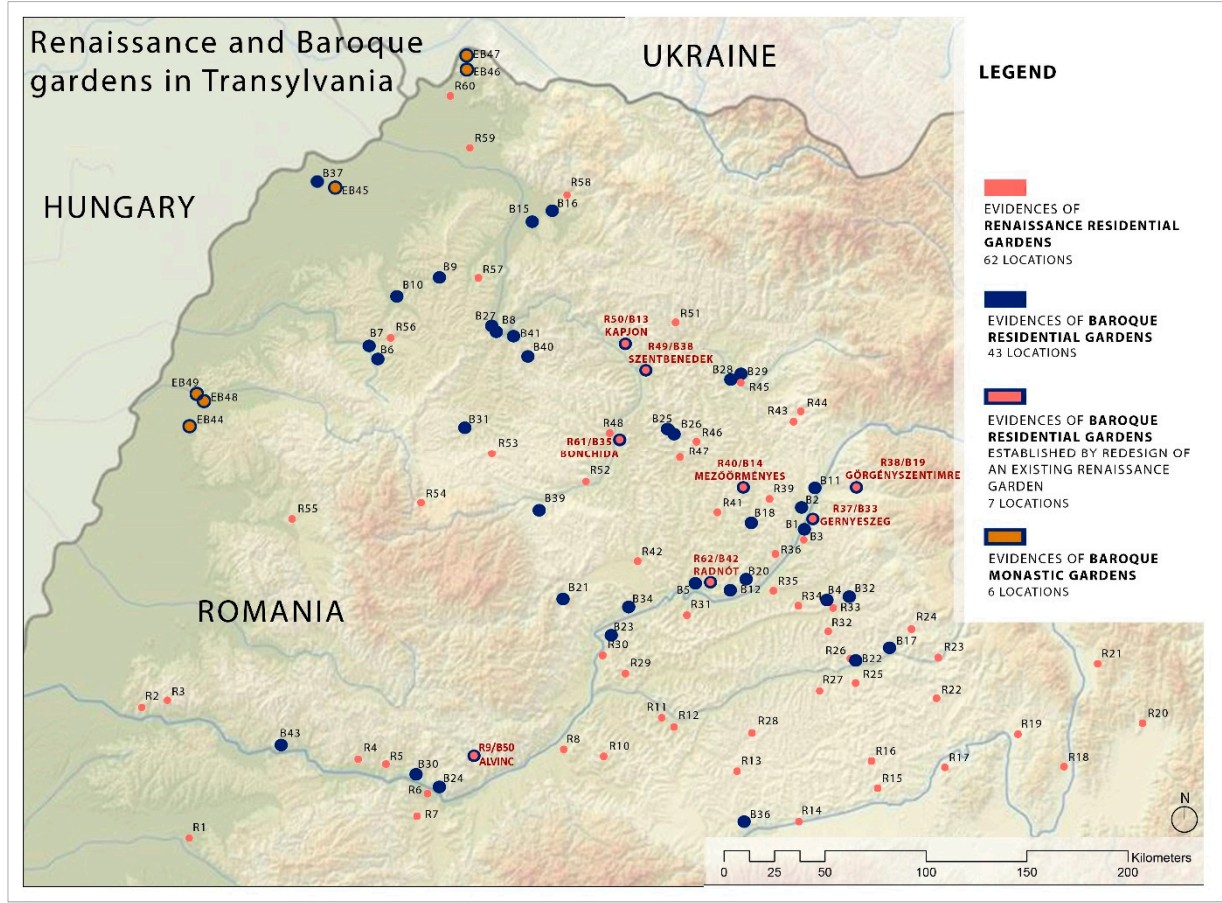

**Figure 2.** Spatial distribution of Transylvanian Baroque gardens and their relation to the pre-Baroque late-Renaissance Transylvanian garden art; the locations of the Baroque gardens marked on the map can be identified by the numbering assigned to them (B1–B50) using Table 2. Source: prepared by Albert Fekete.

**Table 1.** Comparative analysis of the periods of major garden styles between Western Europe and Transylvania. In the last column the delayed style periods are marked with dark background. Source: prepared by Albert Fekete based on [14,37–40].

| Garden Style | | Western Europe | Transylvania | Comments |
|---|---|---|---|---|
| **EARLY RENAISSANCE G** | | 14–15th century | Mathias Corvinus' age (1443–1490) | SIMILAR |
| **LATE RENAISSANCE G** | | 15–16th century | The "Golden Age" of Transylvania (1613–1690) | **CCA 100-150 YEAR DELAY** |
| **BAROQUE G** | | 17–18 century (from 1630s) | 2nd part of 18th c. | **CCA 100 YEAR DELAY** |
| **ENGLISH LANDSCAPE G** | pictoresque | from 1720s | beg of 19th c. to mid 19th c. | **CONSIDERABLE DELAY** |
| | gardenesque | from 1830s | mid 19th c. | **SMALL DELAY** |
| | romantic | from mid 19th c. | mid 19th c. to 2nd part of 19th c. | **ALMOST SIMILAR** |
| **MODERN G** | early modern, art nouveau | turn of 19/20th c. | turn of 19/20th c. | SIMILAR |
| | modern | from 1930 to 1950 | from 1930 to 1950 | SIMILAR |
| | postmodern | from 1950 to 1970 | from 1950 to 1970 | SIMILAR |
| **CONTEMPORARY G** | | from 1970s | from 1970s | SIMILAR |

### 3.2. Research Methodology

Unfortunately, no Transylvanian garden has survived to this day in its original design. In a few places (e.g., Bonchida/Bonţida, Felek/Avrig) the main Baroque compositional elements of the gardens, such as paths, earthworks, alleys, garden structures, etc., can still be seen, but most of the Transylvanian Baroque gardens have been destroyed.

The research is therefore based on archival material and a review of the literature on Transylvanian Baroque garden art. The methodology of the archival research is known.

However, the archival source material on Transylvanian castle gardens is poor. Much of the material has been destroyed, and detailed research of any existing archival material may take a long time due to the lack of proper filing and accessibility of the archives.

The research methodology of Transylvanian baroque gardens was based on the principle that the sites concerned may, and therefore must, be interpreted in context with the related settlements and landscapes as the only way to understand their historic importance and current value [41–43].

For a systematic survey we established the following theoretical framework:

- Identification of all potential sites;
- Thorough garden history research of the sites;
- General landscape assessment of the present conditions of the sites;
- Survey and assessment of the spatial layout and landscape composition.

The goal of the historic research of primary and secondary sources found (archives, library and museum materials, map and postcard collections, thematic bibliography reviews, internet sources, etc.) is to provide a clear idea of the establishment and development of the gardens. It comprises the role the sites play in the landscape and the urban character and layout, and the landscape scale relationships that served as a basis for the establishment of the manor garden and determined the character of the surrounding landscapes to a great extent. The garden history research also deals with the architectural history of the manor house and the family history of the owners.

The site survey precisely records the actual conditions of each garden (sketches, minutes, GPS coordinates, geodetic surveys, plant inventory, digital photographs, etc.) as well as the valuable existing features that are possible to preserve, and thus it serves as a status report and a basis for comparison for conservation strategies and any future restorations. A topographic map (e.g., land registry map) provides the basis for the survey of the general conditions and valuable landscape features.

The garden analysis based on compositional principles helps to define the most important structural types of Transylvanian Baroque gardens, which can serve as a comparison basis for further garden and landscape historical research.

Regarding the literature, we must rely on the works of two of the most dedicated and well-known personalities of Transylvanian architecture and, accordingly, of Baroque garden art, József Biró and Margit B. Nagy.

József Biró examined the palace and the palace garden as a whole in numerous relevant studies [44–46]. In his main work Transylvanian Castles, published in 1943, he devotes a separate chapter to the historical aspects of castle gardens (including the Baroque estates), as well as to the description of their contemporary condition, making it an important source document [47].

Margit B. Nagy primarily researches the architectural heritage of Transylvania, but in her works [11,37,38] she also publishes and processes inventories from the 17th and 18th centuries that contain valuable references to contemporary garden elements and archival source materials presenting them.

Apart from them, several well-known writers in the professional literature and local historians also evoke some aspects of the Transylvanian Baroque garden art [48–54], however, their works are less comprehensive, mostly referring to a single site or detail, and in several cases they refer to the works of Biró or Nagy.

## 4. Research Results

### 4.1. Baroque Works and Their Masters

The masters of gardens belong to two groups. The first group is made up of the noble families, i.e., the owners of the palaces, who in many cases are directly involved in the design and transformation of parks, gardens or individual garden elements. The owners are the source of the spirit and cultural content necessary to create the genius loci, the identity of the place and to develop the residential gardens. In many places, the owners

have designed the gardens according to their own ideas or have directly influenced their design, so in a sense they can be considered as designers.

The second important group of masters is that of professionals, artists and craftsmen. With the help of archival material and other secondary sources, it has been possible to identify several architects, engineers, stone masons, sculptors, gardeners, hydraulic engineers and other professionals who played an important role in the design, construction and transformation of certain Transylvanian castle gardens or their specific elements (Table 2).

**Table 2.** List of professionals and estate owners contributed to the design of the significant Baroque gardens in Transylvania. Source: prepared by Albert Fekete based on research of archives and the literature [7,12,44,54,55].

| Name of Professional | The Castle Garden Where He Worked | Year of Activity | Owner Family |
|---|---|---|---|
| Blaumann Johann Eberhardt | Bonchida (Bonţida), Bánffy Castle Garden<br>Zsibó (Jibou), Wesselényi Castle Garden<br>Felek (Avrig), Bruckenthal Castle Garden | 1776–1783<br>1780s<br>1784 | Bánffy<br>Wesselényi<br>Bruckenthal |
| Bode György | Nagykároly (Carei), Károlyi Castle Garden | 1790s | Károlyi |
| Böhm Anton | Görgényszentimre (Gurghiu), Rákóczi-Bornemisza Castle Garden | 1790 | Bornemisza |
| Burey Francois | Erdőszentgyörgy (Sângeorgiu de Pădure), Rhédey Castle Garden | from 1800 (36 year long) | Rhédey |
| Damm Florian | Nagykároly (Carei), Károlyi Castle Garden | 1783 | Károlyi |
| Erras Johann Christian | Bonchida (Bonţida), Bánffy Castle Garden | 1750s | Bánffy |
| Gindtner Franz | Hadad (Hodod), Wesselényi Castle Garden | 1770 | Wesselényi |
| Hanek Nicolas Philip | Gernyeszeg (Gorneşti), Teleki Castle Garden | 1783 | Teleki |
| Hartmann Gottfried | Kolozsvár (Cluj-Napoca), Bánffy Palais | from 1774 to 1783 | Bánffy |
| Hoffmann János | Kendilóna (Luna de Jos), Teleki Castle Garden | 1744 | Teleki |
| Jarschel Frantz | Mezőmadaras (Mădăraş), Bethlen Castle Garden | 1784 | Bethlen |
| Kopmann Andreas | Sáromberke (Dumbrăvioara), Teleki Castle Garden | 1782 | Teleki |
| Kováts Sámuel | Zsibó (Jibou), Wesselényi Castle Garden<br>Cege (Ţaga), Wass Ádám Castle Garden<br>Cege (Ţaga), Wass Jenő Castle Garden | turn of 18/19 c. | Wesselényi<br>Wass<br>Wass |
| Leder Josef | Hadad (Hodod), Wesselényi Castle Garden | 1787 | Wesselényi |
| Leerch Ferenc | Görgényszentimre (Gurghiu), Rákóczi-Bornemisza Castle Garden<br>Sáromberke (Dumbrăvioara), Teleki Castle Garden<br>Kendilóna (Luna de Sus), Teleki Castle Garden | 1782<br>1783–1784<br>1780s | Bornemisza<br>Teleki<br>Teleki |
| Luidor Jean | Koronka (Corunca), Toldalaghi Castle Garden | after 1770 | Toldalaghi |
| Nachtigall János | Bonchida (Bonţida), Bánffy Castle Garden<br>Zsibó (Jibou), Wesselényi Castle Garden | 1750s<br>1760s | Bánffy<br>Wesselényi |
| Rosenstingl Franz | Nagykároly (Carei), Károlyi Castle Garden | 1783 | Károlyi |
| Schuchbauer Antal | Bonchida (Bonţida), Bánffy Castle Garden | 1750s | Bánffy |
| Serbán Sándor | Gernyeszeg (Gorneşti), Teleki Castle Garden | 1795 | Teleki |
| Überlacher Anton | Hadad (Hodod), Wesselényi Castle Garden | 1790s | Wesselényi |
| Wrabetz Franz | Zsibó (Jibou), Wesselényi Castle Garden | from 1786 to 1800 | Wesselényi |
| "bécsi Inschíner Kapitán" (*engineer from Vienna*) | Bonchida (Bonţida), Bánffy Castle Garden | 1767 | Bánffy |
| "abafáji német kertész" (*German gardener*) | Gernyeszeg (Gorneşti), Teleki Castle Garden | 1792 | Teleki |
| "felső-magyarországi kertész" (*gardener from Upper Hungary*) | Gyulafehérvár (Alba Iulia), Princely Castle Garden | 1680s | Principality of Transylvania |
| "wiener Gartner (der zuvor bei Graf Lacy beschaftigt war)" (*gardener from Vienna, working earlier for Count Lacy*) | Felek (Avrig), Bruckenthal Castle Garden | 1774 | Bruckenthal |

*4.2. Main Types of Baroque Gardens in Transylvania*

　　　Imre Ormos describes the four main types of Baroque gardens, which are defined in the literature on garden history on the basis of the compositional relationship between garden and building [32]. However, the division of the main Baroque garden types is not so obvious. Several interpretations of the main Baroque garden types exist in the literature, which also classify the main Baroque garden types according to a plausible logic based on different classification criteria (e.g., relief, open or closed main axis, main axis perpendicular or parallel to the axis of the palace, etc.) [34,55–64].

　　　In the course of our research on the history of gardens, we have managed to identify a total of 50 sites in Transylvania, where we can assume the presence of Baroque garden art on the basis of mention, detailed description or representation (layout, design, painting, etc.), or can clearly establish it on the basis of surviving garden fragments and garden elements. The majority of the gardens surveyed (forty-three sites) served secular representational purposes associated with the palaces of the aristocracy, while a smaller proportion of the gardens (seven sites) were ecclesiastical (ornamental gardens of religious orders) (Table 3).

**Table 3.** The list of Transylvanian Baroque gardens surveyed, classified by the type of sources used. Source: prepared by Albert Fekete based on research of archives, the literature and site visit.

| Only Mention of Castle Garden (18 Cases) | Detailed Description of Castle Garden (13 Cases) | Detailed Garden Description, Graphic Representation and/or Surviving Garden Features on Site | |
|---|---|---|---|
| | | (12 Cases Castle Gardens) | (7 Cases Monastic Gardens) |
| **B1. Sáromberke** (Dumbrăvioara)—*Teleki* | **B19. Görgényszentimre** (Gurghiu)—*Rákóczi-Bornemisza* | **B32. Erdőszentgyörgy** (Sângeorgiu de Pădure)—*Rhédey* | **EB44. Püspökfürdő** (Băile 1 Mai) —*Nagyváradi Bishop's possession* |
| **B2. Vajdaszentivány** (Voivodeni)—*Zichy* | **B20. Kerelőszentpál** (Chirileu)—*Haller* | **B33. Gernyeszeg** (Gorneşti)—*Teleki* | **EB45. Kaplony** (Căpleni)—*Garden of Franciscan monastery* |
| **B3. Nagyernye** (Ernei)—*Bálintitt* | **B.21. Torockósztgyörgy** (Colţeşti)—*Thoroczkai-Rudnyánszki* | **B34. Marosújvár** (Ocna Mureş)—*Teleki-Mikes* | **EB46. Máramarossziget** (Sighetu Marmaţiei)— *Garden of Piarist monastery* |
| **B4. Kelementelke** (Călimăneşti)—*Simén* | **B22. Fehéregyháza** (Albeşti)—*Haller* | **B35. Bonchida** (Bonţida) —*Bánffy* | **EB47. Bocskó** (Bocicău)— *Parish garden* |
| **B5. Kutyfalva** (Cuci)—*Degenfeld* | **B23. Csombord** (Ciumbrud)—*Kemény* | **B36. Felek** (Avrig)—*Bruckenthal* | **EB48. Nagyvárad** (Oradea)—*Garden of Jesuit monastery* |
| **B6. Szilágybagos** (Boghiş)—*Bánffy* | **B24. Dédács** (Simeria)—*Gyulay–Fáy–Ocskay* | **B37. Nagykároly** (Carei)—*Károlyi* | **EB49. Biharpüspöki** (Episcopia Bihor)— *Bishop's pheasantry* |
| **B7. Szilágynagyfalu** (Nuşfalău)—*Bánffy* | **B25. Cege** (Ţaga)—*Wass György* | **B38. Szentbenedek** (Mănăstireni)—*Kornis* | **EB50. Alvinc** (Vinţul de Jos)—*Garden of Franciscan monastery* |
| **B8. Zsibó** (Jibou)—*Béldi* | **B26. Cege** (Ţaga)—*Wass Ádám* | **B39. Magyarfenes** (Vlaha)—*Jósika* | |
| **B9. Hadad** (Hodod)—*Wesselényi* | **B27. Zsibó** (Jibou)—*Wesselényi* | **B40. Csákigorbó** (Gârbău)—*Haller-Jósika* | |
| **B10. Sarmaság** (Sărmăşag)—*Kemény* | **B28. Kerlés** (Chiraleş)—*Bethlen* | **B41. Szurdok** (Surduc)—*Jósika* | |
| **B11. Abafája** (Apalina)—*Huszár* | **B29. Árokalja** (Arcalia)—*Bethlen* | **B42. Radnót** (Iernut)— *Kornis-Rákóczi-Bethlen* | |
| **B12. Marosugra** (Ogra)—*Haller* | **B30. Déva** (Deva)—*Bethlen* (Magna Curia) | **B43. Soborsin** (Săvârşin)—*Nádasdy-Forray* | |
| **B13. Kapjon** (Coplean)—*Haller* | **B31. Váralmás** (Almaşu)—*Csáky* | | |
| **B14. Mezőörményes** (Urmeniş)—*Rákóczi-Bethlen* | | | |
| **B15. Kővárhosszúfalu** (Satulung)—Teleki | | | |
| **B16. Koltó** (Coltău)—*Teleki* | | | |
| **B17. Fiatfalva** (Filiaşi)—*Ugron* | | | |
| **B18. Mezősámsond** (Şincai)—*Bethlen* | | | |

As we observe in Figure 2, many of the gardens are located close to rivers, which serve not just as water supply sources but as important traffic and transportation corridors as well. The spatial distribution of the baroque gardens follows the location of estate centres, related mainly to urban environments or to main communication and service corridors (roads).

The typological analysis of the Transylvanian Baroque gardens is based on their compositional characteristics, with particular emphasis on the path system and plant composition that determine the layout of the garden. On this basis, I have classified the gardens into five main groups.

### 4.2.1. Gardens with Goosefoot-Pattern Avenues

The goosefoot-pattern avenues are a typical element of Baroque spatial composition and garden layout. It is a combination of avenues running in the extension of the central axis of the main building and in a symmetrical 'V' shape along the axis and terminated by a prominent feature or scene. Although the goosefoot pattern of the avenues can be traced back to the front garden of the Roman Villa, Moltanto at the end of the 16th century, the exemplar of hierarchical organisation of space based on the goosefoot avenues radiating from the main axis of the palace was the gardens of Versailles, which were developed over several decades from the 1660s onwards [34].

However, the goosefoot avenue in France was used not only in Versailles but before in the gardens of Le Notre, starting with Vaux-Le-Vicomte (Richelieu Gardens, 1627). In Versailles it was taken to its maximum expression, and, above all, it was also used for the organisation of urban space.

There are also several earlier examples with radial space compositions outside of France, for instance, Il Tridente di Roma (beginning of the 16th century) or Villa Medicea di Pratolino (Bernardo Buontalenti, 1568). In fact, the radial layout as a mechanism of spatial organisation and landscape definition was used earlier in Spain: Palacio de Aranjuez y las Huertas del Picotajo (Juan bautista de Toledo, 1553). In the Baroque period, this type of layout had a much deeper meaning, as it was usually used to cover the large scales in the forest areas and surrounding territories. It was thus contrasted with the orthogonal layout associated with the palace area. In fact, above both types of layout (radial and orthogonal), a superstructure of compositional axes was superimposed in relation to the axis of the palace. This superstructure dominated the arrangement of the entire layout.

The first simplified Transylvanian version of this classical spatial structure is the Bonchida palace and garden, developed between 1748 and 1752 by Count Dénes Bánffy, which is also the most important Baroque residence in Transylvania [44]. In addition to a pheasantry, the 400-acre estate of Bonchida (Bonţida) also included the most typical Transylvanian Baroque goosefoot-pattern French garden.

From the western facade of the palace, located on the plain of the Small Szamos River, three radial linden alleys extended, each approximately 1000 m long. The starting point of the alleys is the bridgehead built on the main axis of the palace. In addition to the three linden alleys, a fourth shorter horse chestnut alley ran to the north, following the water channel perpendicular to the main axis of the composition [44,47] (Figure 3).

Source: compiled by Albert Fekete, based on Dávid, Gy. [49] and the First Ordnance Survey of Transylvania (1769–1773) [65].

### 4.2.2. Gardens with Mixed (Orthogonal–Goosefoot) Layout

The most typical Transylvanian example of the layout combining elements of both the orthogonal parterres and the goosefoot-pattern walkways is the Bruckenthal Palace Gardens in Felek (Avrig, RO), which was begun to be built between 1761 and 1764 by General Adolf

von Buccow, later governor of Transylvania. After the governor's death in 1764, the estate became the property of Baron Samuel von Bruckenthal, who used it as a summer residence and began to develop the Baroque park of about 6 ha [66]. The ornamental garden, terraced and enclosed by a stone wall because of the considerable differences in levels, is essentially orthogonal in layout, but from the water feature located in the third of the main axis away from the castle, the paths no longer run transversely but five radial walkways start, dividing the southern part of the garden into regular sections (Figure 4).

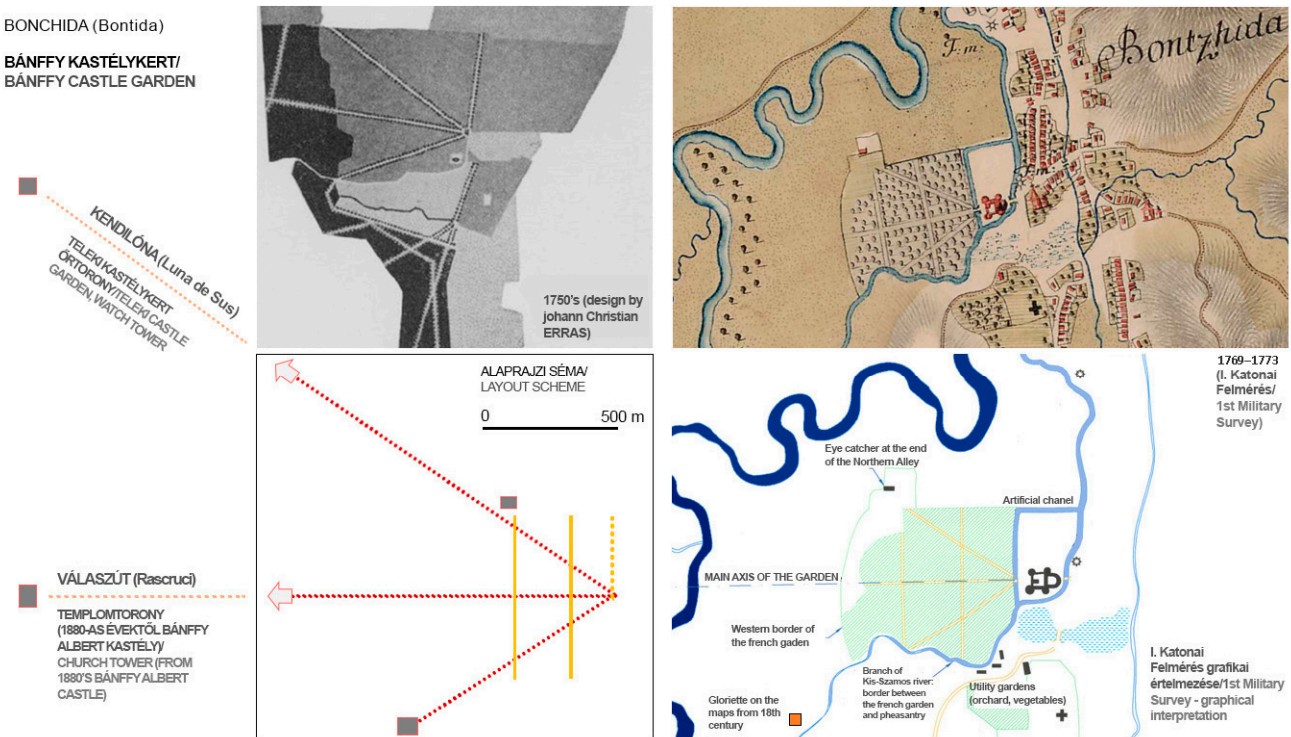

**Figure 3.** Map-based analysis of the goosefoot-pattern Baroque garden in Bonchida.

4.2.3. Gardens with Orthogonal Layout

The most common layout type of Baroque garden in Transylvania is the orthogonal garden. Its prevalence is also due to its relatively simple transparent layout, similar to a Renaissance parterre garden. It consists mostly of square, or occasionally rectangular or irregular square, compartments, sometimes intersected by diagonal or oblique paths. In the composition of the orthogonal gardens, the structural hierarchy typical of the Baroque (e.g., positioning in relation to the main building, side paths subordinate to the main walk) is only rarely present.

Source: compiled by Albert Fekete based on Feyer (2006), the First Ordnance Survey of Transylvania (1769–1773) [61] and "*Mappa prima der neuen Land- und Post-Strassen von Hermanstat bis Cronstat Mappa secunda*" [67]

The subordinate Baroque composition in the hierarchy of the network of paths or the use of the hedges is clearly visible, for example, in the Baroque garden plan of Franz Anton Hillebrandt from 1783 in Nagykároly (Carei, RO) (Károlyi Castle, Figure 5b). The same Baroque gesture of spatial organisation is much more nuanced in the Baroque garden of the Jesuit monastery in Nagyvárad (Oradea, RO), where only the two ornately designed central embroidered parterres emphasise the composition (Figure 5a).

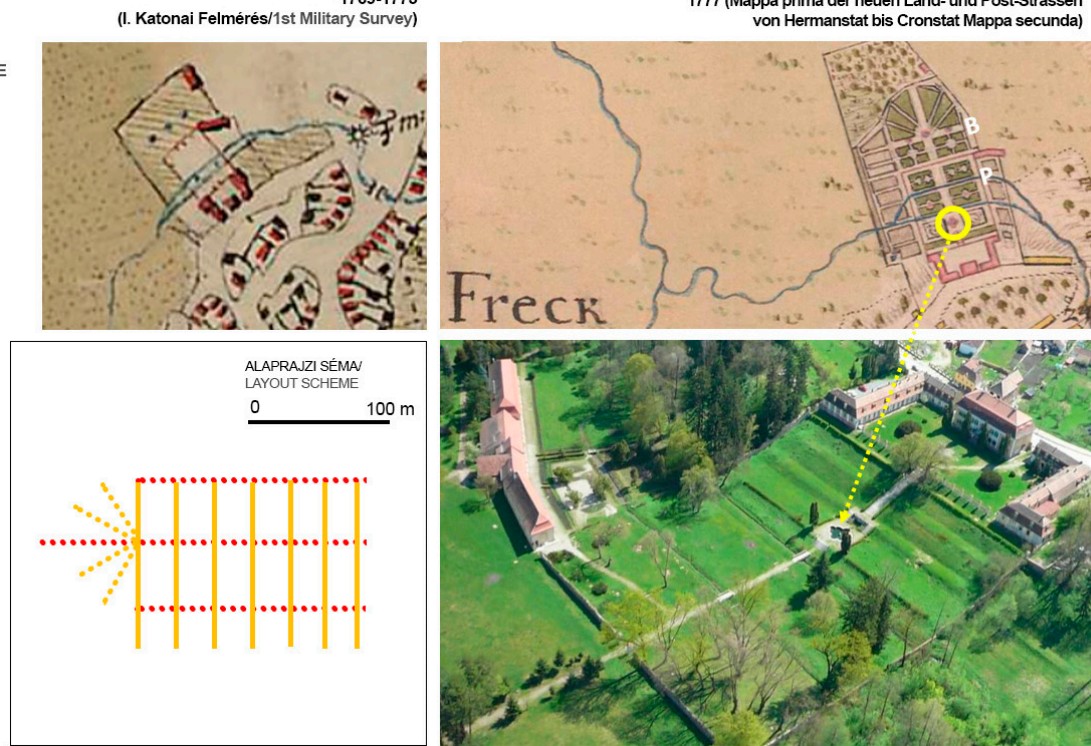

**Figure 4.** Map-based analysis of the mixed-layout Baroque garden in Felek.

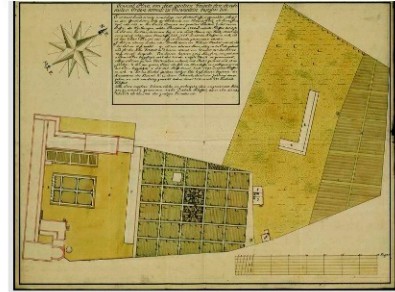

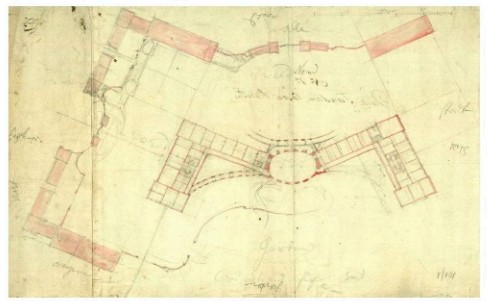

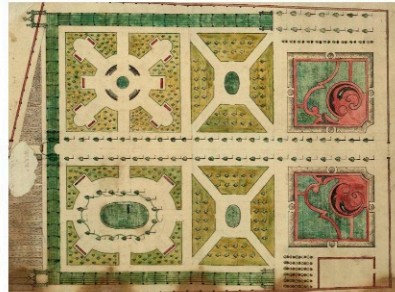

**Figure 5.** Source: compiled by Albert Fekete. (**a**). The Baroque garden plan of Hillebrandt from 1783 for the Károlyi Castle in Nagykároly (Carei, RO) based on "*A tervezett nagykárolyi kastély*" [68]. (**b**). The Baroque garden plan from 1773 for the Jesuit monastery in Nagyvárad (Oradea, RO) based on "*General Plan von dem gantzen Grunde den der Jesuiten Orden vormals zu Groswardein besessen hat*" [69].

The manuscript map of the Rákóczi-Bornemisza Castle in Radnót (Iernut, RO), which was the seat of the prince in the 17th century, shows two very simple orthogonally divided geometric gardens on the west and south-west sides of the castle in the early 19th century.

Neither of these gardens form a Baroque composition with the four-cornered bastion castle building of Renaissance origin.

An aerial photograph showing the current state of the castle and garden shows the outlines and layout of a garden of a similar geometry and size to the gardens on the manuscript map and the ruins of the Baroque gatehouse (Snake House) built on the main southern axis of the castle (the location of the gate is marked by a red circle).

By superimposing the manuscript map and the aerial photograph, it can be seen that the georeferencing does not support the idea that the orthogonal garden (yellow dotted rectangle) shown in the aerial photograph on the main axis connecting the castle and the gate house would have the same size and location as the old garden depicted on the manuscript map (Figure 6).

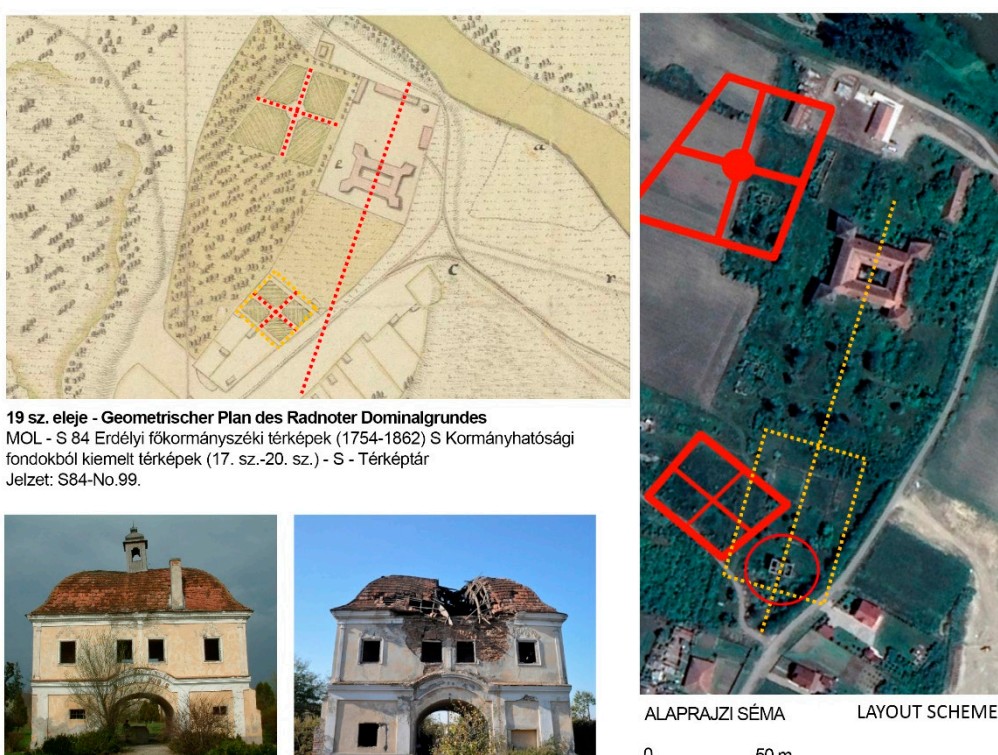

**Figure 6.** Historic analysis of the surroundings of the princely castle in Radnót (Iernut, RO).

Source: compiled by Albert Fekete based on "*Geometrischer Plan des Radnoter Dominalgrundes*" [70], based on a Google Earth image from 2021 and on own photographs.

### 4.2.4. Gardens with Radial Walkways (Star-Shaped Layout)

Radial walkways (star-shaped layout) are a typical compositional feature of the Baroque garden layout, with walkways spreading symmetrically around a central feature (structure, fountain, square). The focal point of the walkways may be located both near to the castle or at a greater distance [60–64].

Instances of radial walkways located near the castle are in Nagykároly (Carei, RO; Figure 7b), Soborsin (Săvârşin, RO; Figure 8) or in Alvinc (Vinţul de Jos, RO). Franz Rosenstingl's 1789 plan of the castle garden in Nagykároly shows a square ornamental garden fitted to the building, with a completely symmetrical layout divided into eight sections with a parterre (or radial walkway) in the middle, which is supplemented by further geometric ornamental garden sections along a garden axis parallel to the longitudinal axis of the castle. The plan shows a subtle redesign of the Baroque garden depicted in the first military survey.

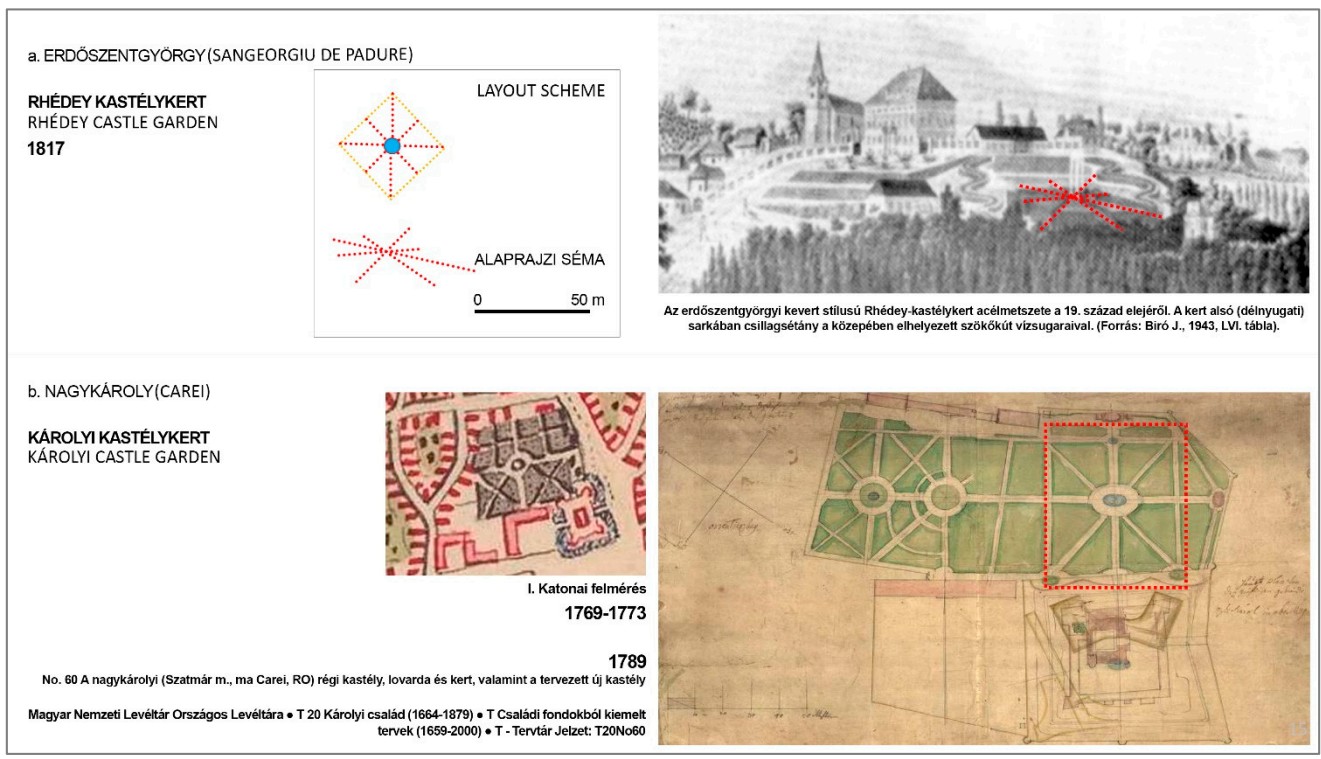

**Figure 7.** Source: compiled by Albert Fekete. (**a**). Steel engraving of the Rhédey castle garden in Erdőszentgyörgy from 1817 [47]. (**b**). Franz Rosenstingl's 1789 plan of the castle garden in Nagykároly (Carei, RO) based on "A nagykárolyi (Szatmár m., ma Carei, RO) régi kastély, lovarda és kert, valamint a tervezett új kastély" [71].

In the case of the castle garden in Alvinc (Vinţul de Jos, RO), a feature of radially shaped walkways in the ornamental garden is mentioned in an inventory from 1676, that is, from the late Renaissance period in Transylvania: "*The ornamental garden was spread out beyond the moat, east of the bridge. Divided by paths in a radiating shape, the flower and vegetable compartments enclosed a gazebo built over the fishpond, which was the centre of an obviously symmetrical composition.*" [37].

A good example of the radial walkways set further away from the castle building is the garden of the Rhédey Castle in Erdőszentgyörgy (Sângeorgiu de Pădure, RO; Figure 7a). It is special due to its "mixed" character, which supports Anna Zádor's observation that 19th century English gardens in Transylvania "*often retained something of the previous garden design, thus showing a somewhat mixed style, adhering to the old and the traditional*" [72], as happened in many renaissance–baroque, baroque–landscape and landscape–Victorian gardens.

From the castle in Erdőszentgyörgy (Sângeorgiu de Pădure, RO) a staircase runs down the terraced hillside to the lower garden and at the bottom of the hill is the bosque, divided by radial walkways with a fountain in the middle shooting high water jets [47].

Source: compiled by Albert Fekete based on "Situations Plan Der Marosch Fluss Laage bey Valliemare und jenseithts/!/bey Soborsin" [73] and own photographs.

### 4.2.5. Baroque Gardens of Local Character

We have descriptions and depictions of several gardens in Transylvania that present Baroque gardens of a specific layout. Among these, one of the most characteristic and unique is the design of the Teleki Castle Garden in Gernyeszeg (Gorneşti, RO), dating from the time of Count József Teleki (I) (Figure 9).

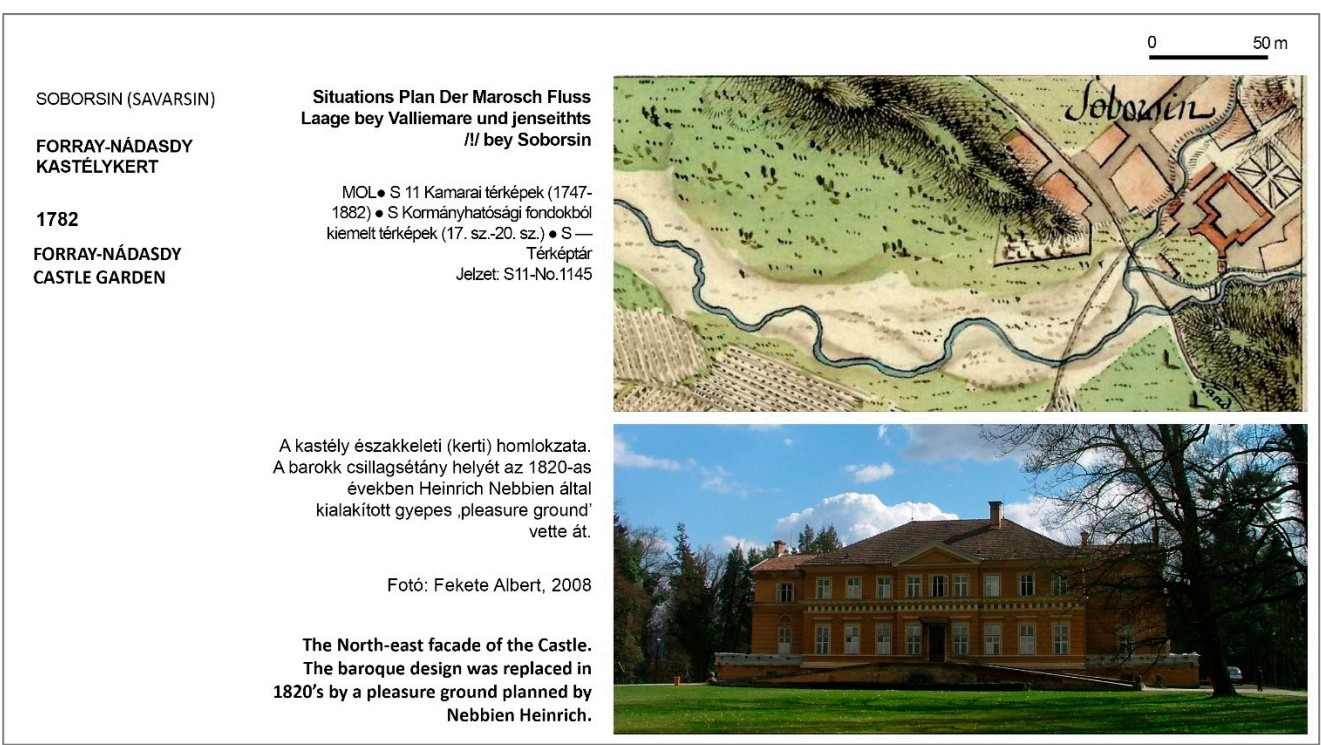

**Figure 8.** The Forray–Nádasdy Castle and its garden in Soborsin (Săvârşin, RO) in 1782.

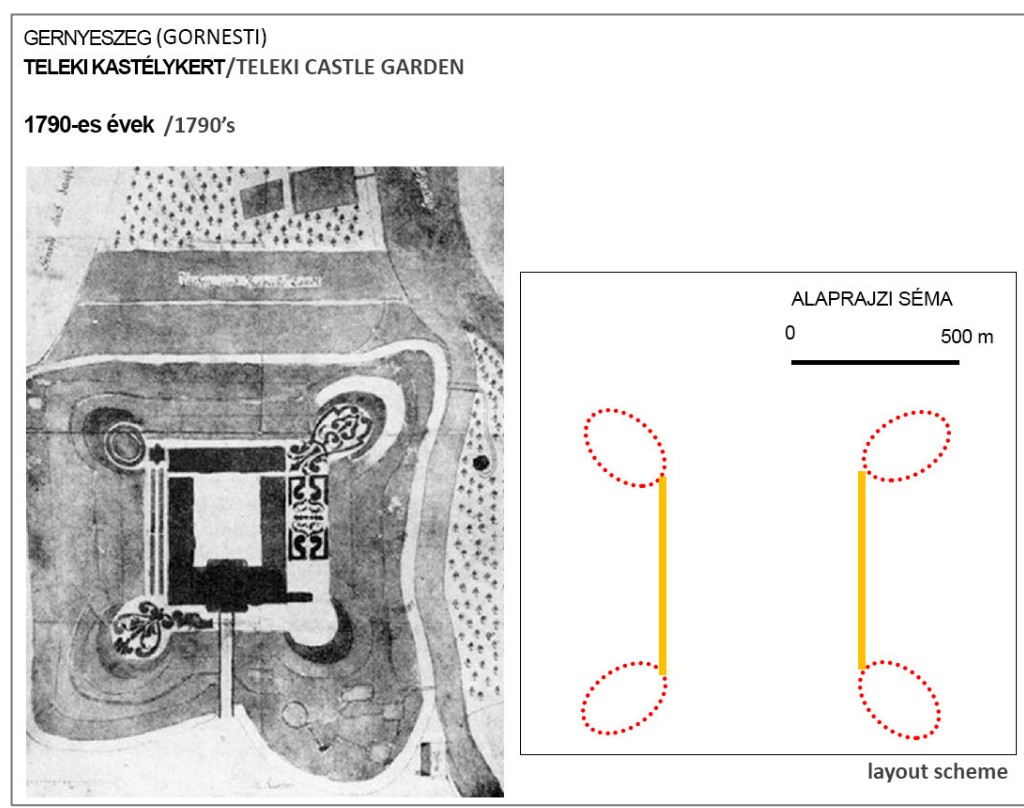

**Figure 9.** The Baroque garden design for the Teleki Castle Garden in Gernyeszeg (Gornesti, RO) from 1792. Source: compiled by Albert Fekete based on "*A gernyeszegi francia kert tervrajza*" [33].

The count's fascination with gardens is well known. During his youthful travels in Western Europe, he visited the famous gardens of his time and, like all works of art, he

showed a keen interest in them. Between 1759 and 1761, he travelled through Austria, Bavaria, Switzerland, the Rhineland principalities, the Netherlands and France. In Germany, he visited the gardens of Rastatt, Karlsruhe, Mannheim and Augustusburg in Brühl, and in the Netherlands he visited the castle and park of ten Bosch, among others [41].

In the 1780s he decided to undertake a major garden renovation in Gernyeszeg (Gorneşti, RO), and thus the plan of 1792 by the "German gardener from Abafája" was born, the design of which was inspired by motifs of local folk decoration. Despite the interesting design elements, the count was not entirely impressed by the design, considering it to be lacking the characteristic elements of the landscape gardens that were then fashionable in Europe. [74].

Following the French Baroque garden pattern, water surfaces (the aquatic element was essential in the French Baroque and was in fact a French addition when the classical gardens were imported to France) were popular elements in some of the Transylvanian Baroque castle gardens and their placement in the geometric Baroque composition also led to specific solutions. In addition to the Haller Castle Garden in Fehéregyháza (Albeşti, RO) and the Bethlen Castle Garden in Kerlés (Chirales, RO), there are references in the case of the Thoroczkay–Rudnyánszky Castle Garden in Torockószentgyörgy (Colţeşti, RO) to a large water basin, which could be used as a fish pond, giving the garden a unique character: "*The large canal pond is an old Transylvanian tradition, as the fish pond in the former ornamental garden of the Count's castle in Torockószentgyörgy, for example, is even associated with a legend*" [44].

## 5. Discussions

In the garden history of Transylvania, the Baroque style was almost one hundred years late in spreading compared with other parts of Europe. In terms of the number of gardens in Transylvania, the Baroque cannot be called the leading garden style in this part of the country. The number of Baroque gardens is much lower than the number of earlier Renaissance or later Landscape Gardens (Figure 10). This is a consequence of the forced political and economic dependence of Transylvania in the 18th century.

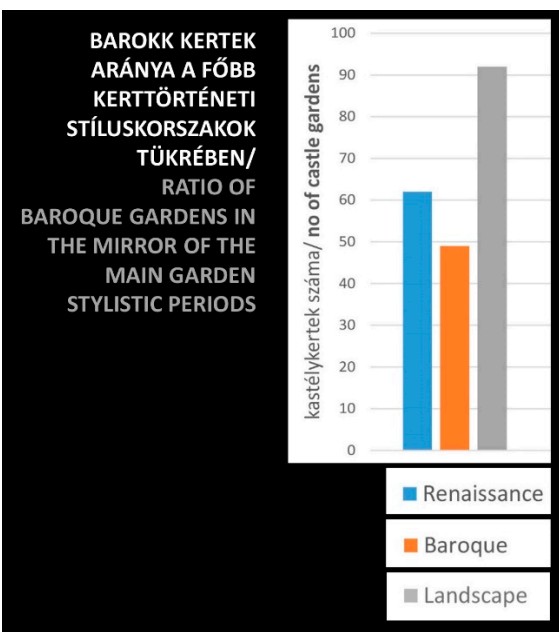

**Figure 10.** The number of Renaissance, Baroque and Landscape Gardens in Transylvania. Source: compiled by Albert Fekete.

The majority of the Baroque gardens realised in Transylvania were of modest design, reflecting the financial constraints of an economically underdeveloped part of the country.

Nevertheless, many foreign masters (gardeners, sculptors, architects, painters, jacks-of-all-trades) of Austrian, German, French, Walloon, Flemish, Czech and Polish origin worked on the design and construction of Baroque gardens of the 18th century Transylvanian noble residences. Their presence proves that Transylvanian Baroque garden art, although geographically marginalised, did not exist in isolation, but was linked to the centre of ideas through its creators and designers. The owners of the estates were aware of the European examples to follow and their artistic qualities.

Somewhat more representative Baroque gardens were developed at the residences of noble families with close ties to the Viennese court and of the dignitaries in the Roman Catholic Church.

The Transylvanian Baroque garden history is mainly about ornamental gardens, although in many places various kinds of kitchen and vegetable gardens or orchards were also part of the Baroque garden. In terms of their compositional characteristics, Transylvanian Baroque ornamental gardens are largely integrated into the general European system. They followed the models, sometimes as simplified small-scale paraphrases of them, but there are also examples that have specific local characteristics (in ethnographic and topographic terms or in plant use).

Given the state of what is left over from these historical artefacts, restoration in the strict sense is almost impossible in the majority of the locations. Devastation, missing archival sources, changing ownerships and sustainability reasons make the restoration work even harder. During the investigation, the analysis and the fieldwork of the Transylvanian ensembles, we had ample contact with local stakeholders, politicians, owners, NGOs, users and other people related to the Transylvanian ensembles. The core of the problem is concentrated around two poles: one of heritage and cultural meaning, the other on the search for new functions and uses. These two are often contradictory and conflicting; they can be categorised in the polarity between development and conservation. In all landscape architectural projects this contradiction plays a role, but in the case of historical phenomena they are even more pronounced and demand special attention [41–43,75].

This will be a major challenge for landscape architecture to take into account the historical values and to integrate them with new functions and use and the recent demands of improving water management, energy transition and the creation of comfort and healthy living environments for people.

## 6. Conclusions

Landscape architecture is an applied science. According to the guidelines on the preparation of scientific publications in garden history, the results of our research are based on general preliminary studies of garden and landscape history, the research results and experiences of several decades, the exploration and analysis of authentic historical sources and the site surveys and assessments. The most important findings of the research are the following:

- **Baroque castle gardens are part of the most significant garden heritage assets in Transylvania.** This finding is supported by the history and the relation to the landscape of dozens of European, Hungarian and Transylvanian castle gardens examined during our preliminary studies and earlier research [33–55], and also by the results of our research based on the survey of 50 Transylvanian baroque gardens introduced in detail in the paper. Due to their compositional characteristics and integration with the surrounding landscape, the baroque gardens have a complex importance as heritage assets.

- **The baroque castle, the garden and the surrounding landscape together represent a single artistic and compositional unit.** Neither the castle nor the garden should be interpreted independently. They are a single unit, and all the man-made and natural elements of the garden and often specific elements beyond the garden boundaries, make part of this unit. The castle and the garden together represent a composition

that is an integral part of a complex system developed on artistic, cultural, historical, ecological and economic bases.

- **Due to the scarce sources in garden history, garden restoration is rarely a feasible option in Transylvania.** In Transylvania, the current conditions of Baroque gardens and the availability of historical sources (often still uncatalogued, inaccessible and incomplete) in most of the cases do not allow for a full restoration of the original design. Nevertheless, the gardens still represent a part of the garden heritage that is possible to restore in the most authentic way on the basis of the features preserved and the data available on the actual conditions.

In summary, it can be said that the Transylvanian Baroque garden is considered an integral part of Hungarian garden history, although its instances are of lower quality than the Hungarian and European averages. Their importance in the field of garden history is due to the fact that Transylvania is the easternmost location of the typical garden styles of early modern Europe, and the Baroque castles and gardens created in Transylvania, mainly on the model of Western Europe, represent some of the easternmost examples of 18th century European garden design.

**Author Contributions:** Conceptualisation, A.F. and M.S.; methodology, A.F.; software, M.S.; validation, A.F. and M.S.; formal analysis, A.F.; investigation, A.F. and M.S.; resources, A.F.; data curation, M.S.; writing—original draft preparation, A.F., writing—review and editing, A.F. and M.S.; visualisation, A.F.; supervision, M.S.; project administration, A.F. All authors have read and agreed to the published version of the manuscript.

**Funding:** This research received no external funding. The publication was supported by the Hungarian University of Agriculture and Life Sciences (MATE).

**Data Availability Statement:** Not applicable.

**Conflicts of Interest:** The authors declare no conflict of interest.

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
