# Peer review of "Baroque Gardens in Transylvania: A Historic Overview"

_land, doi:10.3390/land11060949_

Round 1

Reviewer 1 Report

Thank you for giving me this opportunity to read the manuscript entitled "Formal Gardens in Transylvania. A historic overview". The topic of this manuscript is interesting and would be a good contribution to this field. I think it could be considered for publication in Land once the following issues are addressed.

  1. Please replace the keywords that already appear in the manuscript’s title with close synonyms or other keywords, which will also facilitate your paper to be searched by potential readers.

  1. The “scale” and “compass” are essential elements of a map, and I suggest that the authors add these elements to Figure 1.

  1. I suggest the authors add a “Limitation” as a sub-section to the “Discussion”. A brief discussion of the limitations of the study will help readers better understand the concerns of the article.

  1. The manuscript's language still has some grammatical errors exist (not too much). Therefore, a critical review of the manuscript language will improve readability.

Author Response

Responses to Reviewer no 1

We reworked the text integrating the specific remarks of the reviewers and I have followed the ratings of the criteria: reviewer 1, reviewer 2, reviewer 3, reviewer 4, reviewer 5.

We would like to thank the reviewers for their reviews, suggestions and remarks. It has greatly improved the content, form and text.

Questions/answers:

  • Please replace the keywords that already appear in the manuscript’s title with close synonyms or other keywords, which will also facilitate your paper to be searched by potential readers.

Replaced

  • The “scale” and “compass” are essential elements of a map, and I suggest that the authors add these elements to Figure 1.

Added

  • I suggest the authors add a “Limitation” as a sub-section to the “Discussion”. A brief discussion of the limitations of the study will help readers better understand the concerns of the article.

A new „6. Discussion” section added

  • The manuscript's language still has some grammatical errors exist (not too much). Therefore, a critical review of the manuscript language will improve readability.

Grammatical review done

Reviewer 2 Report

The article presents an interesting and little known case study: the evolution of the classical garden in Transylvania towards compositions that may resonate with Baroque precedents of the 17th and 18th centuries. The gardens studied, usually combined with medieval architecture and elaborated by the nobility of the time, are evidently of lower quality than their European counterparts; nevertheless, their research and comparative study is fully justified as a historical and patrimonial study.

Despite this, the paper has serious shortcomings. In general terms, it reads as a description of pre-elaborated conclusions that prevent proper recognition of the research work. In addition, it suffers from certain deep theoretical flaws that need to be better explained. Therefore, I believe that the authors should rework the paper in both structure and content. Please consider the remarks in the adjoining document as a guide for such reworking.

Author Response

Responses to Reviewer no 2

We reworked the text integrating the specific remarks of the reviewers and I have followed the ratings of the criteria: reviewer 1, reviewer 2, reviewer 3, reviewer 4, reviewer 5.

We would like to thank the reviewers for their reviews, suggestions and remarks. It has greatly improved the content, form and text.

Questions/Answers

  • L 2: Please reconsider the expression "formal gardens", as it implies many more garden styles other than classical Italian and French baroque. For example, the Victorian garden or the American Country Place Era gardens also fall into the "formal" category. Perhaps a better expression would be "Classical Gardens in..."

Reconsidered. The most adequate is: „Baroque garden”

  • L 9-11: Please avoid mentioning specific research groups, departments, or researchers in the

abstract.

Corrected

  • L 14-16: The sentence repeats the ideas of the inner sentence, please rephrase or delete.

Corrected

  • L 23-24: Please:

Replace "garden art" with "History of gardens".

Delete "residential garden" as hardly a castle or palace is identified with residential use in the strict sense of the word.

Delete "bishop garden" as it is not a real part of the article.

Delete "formal garden" (see comments for L2).

Replace "Hungary" with another name that refers to the entity that contained Transylvania at the historical moment in question. Mentioning Hungary may bring confusion with the          current country.

Corrected

  • L 41-43: The authors make repeated references to the fact that we are talking about the easternmost examples of classical garden in Europe (also in L 442-446). This is not exactly so, since there are Russian baroque examples further east (i.e. the gardens of the Peterhof Palace, in St. Petersburg).

Corrected

  • L 46-47: The reference to "civic gardens" feels somewhat odd, since truly civic gardens do not begin to appear in the West until the 19th century (with very few dubious precedents before that).

Deleted

  • L 49-52: As described here, the paper does not fit these objectives and description of the method

Corrected

  • L 55: "2.1. Historical background" should be part of section 1, since it does not talk about materials or methods.

Improved through a reorganization of paper structure and chapter numerotation

  • L 68-69: Please put all the maps at the same scale (a and b are larger without reason) and mark the gardens studied in the place and time where they appear.

Maps resized to the same scale.

The markage of the studied gardens is not recommande at this map scale, because the maps would have be very crowded, hard to understand.

  • L 89-90: Confusing sentence. Please rephrase.

Improved

  • L 96: From here on, the numbering is wrong. This section should remain in section 1.

Reorganised, see answer from Point no. 8

  • L 100-101: Please explain in depth the European baroque style, its most outstanding characteristics and its most recognized examples. To do so, use the new bibliography added (see comment below).

Improved accordingly

  • L 111-112: Join sentence to upper paragraph and quote directly from Fatsar [35].

Done

  • L 114: Please explain in comparison to what. Maybe with French or German baroque?

Explained

  • L 116: Please use a different expression than "the road between western civilization...", as it sounds confusing and unclear.

Corrected

  • L 118-121: Please rephrase.

Done

  • L 123: Please adapt table to journal style and rename as table 1. Equalize information between columns to improve clarity (e.g., always refer to centuries as 1Xth century). Include references in the table itself to justify the periods, as some are very questionable (e.g. 14th century for early Renaissance, since Renaissance culture has a very clear origin in the early 15th century; or gardenesque, which was already a style employed by Repton in suburban houses at the beginning of the 19th century).

Table style adapted to the journal style and renamed accordingly to the requireent of the Reviewer.

Justification with some new included references in the caption of the table.

  • L 127: Here you should begin "2. Materials and Methods".

Already reorganised, see answer from Point no 8

  • L 132-133: Please do not mention methodological issues here if you are talking about sources. These sentences should go later, in their own "Method" section. Also, the sentence "The methodology of the archival research is known" is not valid to explain the method.

Corrected and simplified the chapter subtitle accordingly to the content

  • L 135: Please rephrase "in the storms of history", as it is an expression that does not fit the style of the text.

Corrected

  • L 135-137: If the research was not based on archival material, there is no need to mention it in the text.

The research is partially based on archival materials (as the used figures shows); however we still consider important to mention, that the archival sources are poor and hardly accessible; this is a fact which help to understand some shortcomings of the conclusions

  • L 138: Please replace the expression "proves", since we would already be talking about a conclusion without even having presented the method.

Replaced

  • L 141-142: If the materials used are these, it can be said directly, without the need to justify their use so much.

Done

  • L 142-143: Move to the conclusions section.

Deleted, because is it mentioned in the first paragraph of conclusions

  • L 145-160: The paper should dedicate a specific section to describe in detail what these authors explain, since this information is necessary to contextualize the study cases. It is important, above all, that the authors determine how these references define the Transylvanian Baroque style and how they link it to the European Baroque in general. In addition, many of the references are written in Hungarian, making it difficult for non-Hungarian readers to access this information.

The detailed description of the cited authors’ explanation is a good topic for a next, outsatnding article. We wouldn’t like to enter in details which are not directly linked to the topic of our article.

  • L 161-180: Delete paragraphs and incorporate this information into the table below (the division between owners and professionals). Also make a row for each assignment and include the location of the project. Please check why the last four rows are different.

Owners incorporated in the table 1, see last (additional) column.

The last for rows are different, because the sources doesn’t mentioning the name of the masters, only some datas regarding to their prpofessional background.

  • L 186-192: Please elaborate much more on the explanation of [33, 35, 56-60] and what compositional principles they detect and how they classify gardens. This may repair the fact that, right now, the paper does not have a proper state of the art.

The cited sources gives a quite detailed explanation and classification of the baroques gardens. No sense in our opinion to repeate them.

  • L 193-onwards: Here appears the main problem of the article. First of all, the method is not properly described (What sources are selected? Where do they come from?

Improved accordingly

  • How are the graphic documents treated (catalogued, vectorized, overlaid, drawn over...)? From what is presented in the paper we observe very diverse operations. For example, Figure 6 compares two maps with a vectorial reconstruction apparently elaborated by the authors. While figure 7 shows a similar process, figure 9 shows drawings made directly on old cartography and current satellite images. Figure 10, for example, shows a combination of the above. All of this needs to be properly clarified.

Each figure has the sources marked in the caption.

  • Once the method has been described, the results have to be presented. The types described in subsections 5.1 onwards are really conclusions. It is necessary to include in the paper a systematic view of all the gardens studied, either all the maps placed in a matrix or all the vectorized plans following the same graphic code. Then, a comparative reading of this information must follow. From there you will derive a typological classification (or several).

The diversity of graphical representations is coming from the different image qualities (sometimes with relative low resolutions) of the sources available. The different types of graphical helps to offer a readabel and un understandable quality of the materials presented in the article.

  • L 198: Please delete this sentence as it is understood that the research is oriented towards palace gardens.

Sentence deleted

  • L 207: Delete this sentence and insert a direct reference to this figure in the text.

Sentence deleted, direct reference to figure 4 included in the text.

  • L 210: Figure 4 deserves much more analysis in the text. It is convenient to value the fact that most of the gardens are close to rivers, to determine if they were placed near urban environments or if they were related to some type of infrastructure of communication or service.

Completed as requested.

  • L 215: Replace "typology" with "typological analysis". Also, in the description of the method, it would be helpful to explain why the analysis is based on composition.

Improved accordingly

  • L 222: Please delete figure as it is confusing.

Figure 5 deleted

  • L 227-230: In the baroque period, this type of layout had a much deeper meaning, as it was usually used to cover the large scales in the forest areas and surrounding territories. It was thus contrasted with the orthogonal layout, associated with the palace area. In fact, above both types of layout (radial and orthogonal), a superstructure of compositional axes was superimposed in relation to the axis of the palace. This superstructure dominated the arrangement of the entire layout.

Included in the text

  • L 231: There are several earlier examples. To name a few: Il Tridente di Roma (Beginng 16th century); Villa Farnese (Vignola, c1560), or Villa Medicea di Pratolino (Bernardo Buontalenti, 1568). In fact, the radial layout as a mechanism of spatial organization and landscape definition was used earlier in Spain: Palacio de Aranjuez y las Huertas del Picotajo (Juan bautista de Toledo, 1553).

Included in the text (excepting Villa Farnese, which has an ortogonal layout)

  • L 233: Not only Versailles, but in the gardens of Le Notre, starting with Vaux-Le-Vicomte. Although in France it was already used before (Richelieu Gardens, 1627). In Versailles it is taken to its maximum expression, and, above all, it is also used for the organization of urban space.

Included in the text

  • L 245: Please remove, both here and in all figures, the Hungarian text.

We consider the bilingual text important; it is used to show some original, often archaic wording which can’t be translated and which can contribute to an authentical interpretation

  • L 252-253: This sentence supposes conclusions and, moreover, not very accurate. The combination of layouts is very characteristic of the baroque garden. In the article, orthogonal layout is being confused with Renaissance layout. The orthogonal layout was used much earlier. What we see here is baroque layout, i.e. orthogonal combined with radial in a systematic way.

Revised

  • L 257-262: This example is much more closely related to the Italian terraced garden (i.e. Villa Lante). The fact that it was design at a certain date or that it includes a radial layout does not mean that it is baroque, but classical.

It is question of interpretation

  • L 264: The term "baroque" here is misapplied. It seems to be no more than a decorative garden, without any truly baroque attributes such as being an expression of absolute power, seeking the fictitious capture of infinity, extension to scales of the landscape, total concealment of the undesigned environment, creation of spatial anamorphisms, and so on.

In many cases some small residential gardens, belonging to less powerfull owners follows also baroque compositional rules; many Transylvanian gardens falls in this section; despite these doesn’t representing at the first sight the absolute power of the aristocracy, still can be considered as modest representants of this garden style.

  • L 281-282: The parterre is a baroque element, but it does not make a baroque garden. In this case, the paper is studying the reinterpretation of classical elements.

It is question of interpretation

  • L 304-308: Please, clarify this. It changes the method and also discusses an idea that has not been postulated before.

Reedited

  • L 318-321: Please include bibliographical references.

References included

  • L 348: As happened in many renaissance-baroque; baroque-landscape and landscape-victorian gardens.

Observation included in the text.

  • L 361: "Less conventional layout" according to what specific author?

Changed into: „specific”

  • L 364-372: Unnecessary information.

Reduced

  • L 384: The aquatic element was essential in the French Baroque and was in fact a French addition (the large body of water along extensive forests) when the classical gardens were imported to France.

Observation included in the text.

  • L 394: It is absolutely necessary that before the conclusions there be an extensive discussion section in which the results are contrasted with the state of the art.

Discussion (chapter 6) has been inserted and article has been extended accordingly.

  • L 399-400: This idea should be reinforced with bibliographical references.

New references included.

  • L 406: Be careful with the notion of "modest design" since it seems to contradict the baroque spirit itself.

Revised accordingly

  • L 413: At some point in the paper, the transfer of these foreign ideas should be commented on more precisely.

See new included literature and references

  • L 415-417: Which ones?

Revised

  • L 421-424: These ideas deserve more study throughout the paper.

In our vision, the presented examples are proving this ideas.

  • L 430-433: this needs to be discussed.

Supported by some new related references

  • L 469: Adapt reference to style.

Adapted

  • L 493-496: Delete reference and use a more accessible one.

Improved accordingly

-     L 501-515: Please include and discuss some of the classic books on classical and baroque garden art: i.e. La Cattura dell'Infinito (Benevolo, 1991); The Italian Renaissance Garden (Lazzaro, 1990); Gartenkunst der Renaissance und des Barock (Hansmann, 1983); Gardens of Illusion: The Genius of André Le Notre (Hamilton Hazlehurst, 1980); The French Garden 1500-1800 (Adams, 1979).

Improved accordingly

Reviewer 3 Report

Dears, I have serious doubts about the Originality and Scientific Soundness of this publication.

It is written well, like a novel. You even find time to show us the five biggest patrons of Baroque gardens in Transylvania (fig 3), but I can not find:

-comparison to other gardens

-advice on how other gardens should be mainteince

-suggestion what should be changed in this garden

The title suggests a historical story, but I am not sure if this magazine is a place for historical stories.

I wish you all the best

Author Response

Responses to Reviewer no 3

We reworked the text integrating the specific remarks of the reviewers and I have followed the ratings of the criteria: reviewer 1, reviewer 2, reviewer 3, reviewer 4, reviewer 5.

We would like to thank the reviewers for their reviews, suggestions and remarks. It has greatly improved the content, form and text.

Questions/Answers

- It is written well, like a novel. You even find time to show us the five biggest patrons of Baroque gardens in Transylvania (fig 3), but I can not find:

- comparison to other gardens

  • Meanwhile - during the review process - we included and mentioned some new examples from abroad as well, for instance some early baroque compositions like Il Tridente di Roma, Villa Medicea di Pratolino, Palacio de Aranjuez y las Huertas del Picotajo, Vaux-Le-Vicomte, Hampton Court, Het Loo, Frederiksborg, Herrenhausen, Nymphenburg, Schönbrunn, Peterhof

  • advice on how other gardens should be mainteince

We do not consider the details related to maintanence of the gardens as part of this topic; that’s why we wouldn’t deal with this aspect (it can be another, outstanding article)

  • suggestion what should be changed in this garden

We do not consider the details related to renewal design and proposal of the gardens as part of this topic; that’s why we wouldn’t deal with this aspect (it can be another, outstanding article); see some general related considerations in the last paragraph of the Conclusions

  • The title suggests a historical story, but I am not sure if this magazine is a place for historical stories.

We consider the topic fit in the topic of this special issue; the editors of the special issues agreed it also 

Reviewer 4 Report

This article is important for a systematic review of Transylvanian gardens. Against this background, I have only a few minor observations to make:

  1. The text might benefit from an additional round of proofreading. There are still some minor spelling mistakes left (e.g., "Transylvanian ensembles, gardens and parks have been investigated, described and analysed by a research group from [the] Institute of Landscape Architecture [...]" (R8-R9), or "leadby Albert Fekete", instead of "lead by Albert Fekete" (R11)).
  2. The authors should recheck the spelling of the historical text passages (e.g., in Figure 8a, the illustration reads "[...] zu Grosswardein bessen [?] [...]", instead of "[...] zu Grosswardein besessen [...]" as it reads in the description of Figure 8b (R301). Please check also the article in the title of Figure 11: shouldn't it read "Situations Plan des Marosch Flusses [...]", instead of "Situations Plan der [?] Marosh Fluss" (R352-R357).
  3. Diacritical marks for all Romanian spellings are missing (e.g., "Bontida" should read "Bonțida", etc.), but they appear in the Hungarian spellings. Was this intentional?
  4. What does the final (coloured) column in Figure 2 show (R122-R123)? What do the colours mean?
  5. I did not quite get what Ormos's typology consists of. Does it reflect in the ensuing classification? A short description of his classification might help the reader grasp the authors' own classification better.
  6. Table 2 seems to have some background colour on the headings of the last two columns. In addition, I would personally align all text in the tables on the left.
  7. And finally, the abstract mentions restoration efforts ("The conclusions is [are] that, given the state of what is left over from these historical artefacts, restoration in the strict sense will be impossible. This will be a major challenge for landscape architecture to take into account the historical values, integrate them with new functions, use and the recent demands of improving water management, energy transition and the creation of comfort and healthy living environments for people" (R18-R22). However, the article only mentions possible restoration works only in the concluding section. This seems to be only an add-on, so I would either delve deeper in restoration matters, i.e., dedicate a special section to it, or just leave the restoration part out and stick to the classification, which is in itself very helpful for Transylvanian garden history research.

Other than that, the article is easy to understand and provides the (interested) reader with some valuable typological information.

Author Response

Responses to Reviewer no 4.

We reworked the text integrating the specific remarks of the reviewers and I have followed the ratings of the criteria: reviewer 1, reviewer 2, reviewer 3, reviewer 4, reviewer 5.

We would like to thank the reviewers for their reviews, suggestions and remarks. It has greatly improved the content, form and text.

Questions/Answers

This article is important for a systematic review of Transylvanian gardens. Against this background, I have only a few minor observations to make:

  • The text might benefit from an additional round of proofreading. There are still some minorspelling mistakes left (e.g., "Transylvanian ensembles, gardens and parks have beeninvestigated, described and analysed by a research group from [the] Institute of LandscapeArchitecture [...]" (R8-R9), or "leadby Albert Fekete", instead of "lead by Albert Fekete"(R11). Corrected

  • The authors should recheck the spelling of the historical text passages (e.g., in Figure 8a, theillustration reads "[...] zu Grosswardein bessen [?] [...]", instead of "[...] zu Grosswardeinbesessen [...]" as it reads in the description of Figure 8b (R301). Please check also the articlein the title of Figure 11: shouldn't it read "Situations Plan des Marosch Flusses [...]", insteadof "Situations Plan der [?] Marosh Fluss" (R352-R357).
  •  
  • The grammatical problems and spelling you mention has its origins in the archaic German used at that time (during the 18/19Th Centuries). We reproduced in our references exactly the same text we found on the archival materials.
    • - Diacritical marks for all Romanian spellings are missing (e.g., "Bontida" should read"BonÈ›ida", etc.), but they appear in the Hungarian spellings. Was this intentional?
  • We corrected. In case of all Transylvanian settlement we gave the Hungarian and Romanian names written according to the language-specific diacritical marks.
  • What does the final (coloured) column in Figure 2 show (R122-R123)? What do the coloursmean  
  •  The table has been reedited, we hope now is more understandable.
  • - I did not quite get what Ormos's typology consists of. Does it reflect in the ensuing classification? A short description of his classification might help the reader grasp the authors' own classification better.
  • It is detailed in the given literature
  • - Table 2 seems to have some background colour on the headings of the last two columns. In addition, I would personally align all text in the tables on the left.
  • Improved accordingly.
  • And finally, the abstract mentions restoration efforts ("The conclusions is [are] that, given the state of what is left over from these historical artefacts, restoration in the strict sense will be impossible. This will be a major challenge for landscape architecture to take into account thehistorical values, integrate them with new functions, use and the recent demands ofimproving water management, energy transition and the creation of comfort and healthyliving environments for people" (R18-R22). However, the article only mentions possible restoration works only in the concluding section. This seems to be only an add-on, so I would either delve deeper in restoration matters, i.e., dedicate a special section to it, or just leavethe restoration part out and stick to the classification, which is in itself very helpful forTransylvanian garden history research.
  • We do not consider the details related to renewal design and proposal of the gardens as part of this topic; that’s why we wouldn’t deal with this aspect (it can be another, outstanding article); see some general related considerations in the last paragraph of the Conclusions
  • Other than that, the article is easy to understand and provides the (interested) reader with some valuable typological information.

Reviewer 5 Report

Dear Author(s),

The article is interesting and the study brings up some important issues.

However, the manuscript requires some improvements.

The authors should mention in the key words that Transylvania is a region located in Romania, not in Hungary, in order not to create confusion (fig. 4 specifies the position of the region).

The aim and objectives of the paper should be clearly stated at the end of the introduction.

The introduction should briefly mention the findings from the literature in relation to the research topic.

The paper should be structured in parts that clearly highlight the research approach: introduction, methods, research results , conclusions.

Pay attention to the numbering of the chapters in the paper (example : 2. Materials and methods, 2.1 Historical background , then again 2.  Timelines of key European garden styles, 3. ....

A cartographic illustration of the distribution of the main types of Baroque gardens in Transylvania (gardens with goosefoot-pattern avenues, gardens with mixed (orthogonal-goosefoot) layouts....) is an addition to the paper (chapter 5).

The conclusions should point to the strengths and weaknesses of the conducted analyses, suggest further directions of research.

Best regards.

Author Response

Responses to Reviewer no 5.

We reworked the text integrating the specific remarks of the reviewers and I have followed the ratings of the criteria: reviewer 1, reviewer 2, reviewer 3, reviewer 4, reviewer 5.

We would like to thank the reviewers for their reviews, suggestions and remarks. It has greatly improved the content, form and text.

Questions/Answers

The article is interesting and the study brings up some important issues. However, the manuscript requires some improvements.

The authors should mention in the key words that Transylvania is a region located in Romania, not in Hungary, in order not to create confusion (fig. 4 specifies the position of the region).

Included

The aim and objectives of the paper should be clearly stated at the end of the introduction.

Improved

The introduction should briefly mention the findings from the literature in relation to the research topic.

It is included

The paper should be structured in parts that clearly highlight the research approac: introduction, methods, research results, conclusions.

Improved accordingly

Pay attention to the numbering of the chapters in the paper (example: 2. Materials and methods,2.1 Historical background, then again 2. Timelines of key European garden styles, 3. ....

Article reedited and chapters renumbered

A cartographic illustration of the distribution of the main types of Baroque gardens inTransylvania (gardens with goosefoot-pattern avenues, gardens with mixed (orthogonal-goosefoot) layouts....) is an addition to the paper (chapter 5).

OK

The conclusions should point to the strengths and weaknesses of the conducted analyses, suggest further directions of research.

Included

Round 2

Reviewer 2 Report

I am grateful to the authors for having included some of the modifications presented in the review. Even so, they have not implemented the substantial changes that would imply a real modification of the article, making it suitable for publication in a prestigious journal such as Land. Therefore, I maintain that the best option for the authors is to withdraw the article, modify it substantially and send it again for review. Several of the relevant issues are discussed in the attached file.

Author Response

Dear Reviewer,

As far as we received five reviews, during the rework process of the original text we tried to follow all the recommandations of the reviewers. It seems their opinions doesn’t overlap hundred procent: according to the reviews, our paper has been fully accepted by the Reviewer 1, Reviewer 3, Reviewer 4 and Reviewer 5.

Only the Reviewer 2 is still considering necessary modifications in the article. Following the 2nd Reviewer’s comments and recommendations, the paper would change considerable its content, which already has been accepted by the others.

Reviewer 3 Report

The authors have determined that the content and title fit the special issue and have the editors' approval, I should not question the title.

Thank you for addressing my suggestions. I accept all replies and propose to publish them as follows 

Author Response

Dear Reviewer,

Thank you for your acceptance

Reviewer 5 Report

Dear Author (s),

I'm glad you took into account the suggestions I made.

However, the manuscript still requires minor improvements for clarity.

  1. The text dealing with issues related to the methodology of the paper (lines 200 -225) should be moved to sub-chapter 3.2 - research methodology. The research methodology should be included in one chapter, regardless of the type of analysis/classification of the gardens. Chapters 4 and 5 represent the research results (analysis of different types of gardens)
  2. Suggestion: The paper to be structured in : 1.Introduction, 2. Historical background , 3. Materials and methods. 4. Research results (chapters 4 and 5 will become subchapters of this chapter), 5. Discussion and 6. Conclusions.
  3. minor editing errors to be corrected: empty columns in Table 2 (rows 176-177) should be deleted. The same situation occurs in Table 3, rows 196-197)

Author Response

The text dealing with issues related to the methodology of the paper (lines 200 -225) shouldbe moved to sub-chapter 3.2 - research methodology. The research methodology should beincluded in one chapter, regardless of the type of analysis/classification of the gardens. Chapters 4 and 5 represent the research results (analysis of different types of gardens)

Suggestion: The paper to be structured in: 1.Introduction, 2. Historical background , 3.Materials and methods. 4. Research results (chapters 4 and 5 will become subchapters ofthis chapter), 5. Discussion and 6. Conclusions.

Article restructured accordingly to the suggestions

Minor editing errors to be corrected: empty columns in Table 2 (rows 176-177) should bedeleted. The same situation occurs in Table 3, rows 196-197)

            Indicated errors corrected